# Evaluation of the Synthetic Multifunctional Peptide Hp-MAP3 Derivative of Temporin-PTa

**DOI:** 10.3390/toxins15010042

**Published:** 2023-01-05

**Authors:** Patrícia Souza e Silva, Alexya Sandim Guindo, Pedro Henrique Cardoso Oliveira, Luiz Filipe Ramalho Nunes de Moraes, Ana Paula de Araújo Boleti, Marcos Antonio Ferreira, Caio Fernando Ramalho de Oliveira, Maria Ligia Rodrigues Macedo, Luana Rossato, Simone Simionatto, Ludovico Migliolo

**Affiliations:** 1S-Inova Biotech, Postgraduate Program in Biotechnology, Universidade Católica Dom Bosco, Campo Grande 79117-900, Mato Grosso do Sul, Brazil; 2Laboratório de Purificação de Proteínas e suas Funções Biológicas, Unidade de Tecnologia de Alimentos e da Saúde Pública, Universidade Federal de Mato Grosso do Sul, Campo Grande 79070-900, Mato Grosso do Sul, Brazil; 3Laboratório de Pesquisa em Ciências da Saúde, Universidade Federal da Grande Dourados UFGD, Dourados 79825-070, Mato Grosso do Sul, Brazil

**Keywords:** bacterial resistance, fungal infections, antimicrobial peptides, rational design, temporins

## Abstract

In recent years, antimicrobial peptides isolated from amphibian toxins have gained attention as new multifunctional drugs interacting with different molecular targets. We aimed to rationally design a new peptide from temporin-PTa. Hp-MAP3 (NH_2_-LLKKVLALLKKVL-COOH), net charge (+4), hydrophobicity (0.69), the content of hydrophobic residues (69%), and hydrophobic moment (0.73). For the construction of the analog peptide, the physicochemical characteristics were reorganized into hydrophilic and hydrophobic residues with the addition of lysines and leucines. The minimum inhibitory concentration was 2.7 to 43 μM against the growth of Gram-negative and positive bacteria, and the potential for biofilm eradication was 173.2 μM. Within 20 min, the peptide Hp-MAP3 (10.8 μM) prompted 100% of the damage to *E. coli* cells. At 43.3 μM, eliminated 100% of *S. aureus* within 5 min. The effects against yeast species of the *Candida* genus ranged from 5.4 to 86.6 μM. Hp-MAP3 presents cytotoxic activity against tumor HeLa at a concentration of 21.6 μM with an IC_50_ of 10.4 µM. Furthermore, the peptide showed hemolytic activity against murine erythrocytes. Structural studies carried out by circular dichroism showed that Hp-MAP3, while in the presence of 50% trifluoroethanol or SDS, an α-helix secondary structure. Finally, Amphipathic Hp-MAP3 building an important model for the design of new multifunctional molecules.

## 1. Introduction

A large number of infectious diseases have become treatable and curable due to the discovery and development of antimicrobial agents. However, today, some 90 years after the discovery of penicillin, the threat of bacterial and fungal infections has become a major concern worldwide [1]. Bacteria from the ESKAPE group (*Enterococcus faecium*, *Staphylococcus aureus*, *Klebsiella pneumoniae*, *Acinetobacter baumannii*, *Pseudomonas aeruginosa*, and *Enterobacter*) resistant to antimicrobials are making treatment options even more scarce, due to the high rate of resistance to the last line of antibiotics on the market [2].

The most prevalent opportunistic fungal pathogens worldwide are of the *Candida* genus, microorganisms that colonize the skin, oral cavities, gastrointestinal and genital-urinary tract of most healthy people. Dysbiosis in the microbial community and alterations in the immune system may turn species such as C. albicans and other emerging species, including *C. auris, C. glabrata, C. krusei, C. tropicalis*, and *C. parapsilosis*, into pathogens, leading to the emergence of various diseases; from skin infections to severe systolic infections, with a high mortality rate in hospitalized patients [3,4,5]. Systemic infections by *C. albicans* lead to a mortality rate of ~40% [6] and are the main responsible for invasive candidiasis (46.3%), followed by *C. glabrata* (24.4%) and *C. parapsilosis* (8.1%) [7].

These pathogens can form biofilms, which represent an architectural complex cell group aggregated into an extracellular polymeric substance (EPS). Biofilms are intrinsically tolerant of conventional antibiotics. The unique phenotypic and metabolic properties of biofilms make it possible to implement resistance mechanisms at the community level, making them up to 1000 times more resistant to antimicrobial agents than their plankton forms. The presence of EPS hinders the permeability of antibiotics in the structure of the biofilm, becoming difficult to eradicate it. The occurrence of a gradient of permeability also creates subpopulations of cells in a dormant (persistent) state with low metabolic activity [8,9].

These infections become more aggressive in cancer patients, as these pathogens are significantly associated with higher morbidity and mortality rates in these patients. According to a report by the World Health Organization, cancer is responsible for about 10 million deaths worldwide by 2020 [10]. Conventional treatments involving surgery, chemotherapy, and radiotherapy, along with newly developed immunotherapy, have been applied to eliminate cancer cells or inhibit their proliferation [11]. Lung cancer is the leading cause of cancer death worldwide, with 2.21 million cases [12]. Persistent high-risk HPV infections are the main etiological factor of cervical cancer, with 1.4 million cases worldwide [13,14,15]. This type of cancer may lead to changes in immunity and the vaginal microbiota, making the patient more susceptible to resistant bacterial and fungal infections [16]. Rhabdomyosarcoma (RD), a soft tissue sarcoma, is responsible for ~7% of cancers in children. They spread mainly sporadically (90%), rarely associated with an underlying constitutional genetic disease (10%). It represents a high-grade neoplasm of skeletal cells similar to myoblasts. The treatment is invasive, by surgical resection and/or ionizing radiation, and eradication of systemic metastatic disease with intensive chemotherapy [17,18,19]. Thus, the discovery of new multifunctional drug therapies against bacteria, fungi, and cancer cells is essential for patients to have new alternatives in the treatment of various diseases.

To overcome the crisis of microbial and tumor cell resistance, the search for small molecules that are active against these different molecular targets is highlighted [20]. Recently AMPs have gained increasing attention as new antimicrobial candidates to combat infections caused by biofilm-forming bacteria, fungi, viruses, protozoa, tumoral cells in culture, and immunomodulators [21,22,23,24,25]. They share the disruption of plasma membrane integrity as the main mechanism of action [26]. AMPs can be obtained from a wide variety of sources, including microorganisms, plants, insects, mollusks, fish, and amphibians [27]. Amphibian skin secretion peptides actively kill microbes and prevent infection naturally [28]. Among the sets of AMPs identified in the skin of anurans are the temporins, a family of AMPs with more than 100 isoforms in total. among the naturally occurring AMPs, these molecules are the shortest, usually containing in their peptide chain from 10 to 14 amino acid residues. Temporins are hydrophobic and amidated at their C-terminus, distinguished by a low cationic charge, ranging from+2 to +3 [29].

In general, temporins show activity only against Gram-positive bacteria with minimum inhibitory concentration (MIC) values between 1 to >100 μM [30]. The most studied temporins are temporin A (TA), temporin B (TB), and temporin L (TL) for the rational design of new molecules [31]. Synthetic temporin-PTa whose physicochemical characteristic is a net charge of +2, hydrophobicity 0.93, and hydrophobic moment of 0.72, is rarely reported in design studies [32]. Even though it has the potential for microbial identification on a biosensor platform [33]. Rational design can be used to modify the chemical and physical properties of existing peptides, thereby improving activity against targets or making them multifunctional for multiple targets [34,35]. Amphipathic peptides rich in lysines and leucines are multifunctional in various pathogenic cells [36,37]. Since these attributes are widely accepted for the design of synthetic AMPs. The objective of this work was the rational design of a new amphipathic peptide rich in lysine and leucine with multifunctional potential against bacteria, fungi, and tumor cells.

## 2. Results

### 2.1. Rational Design and Synthesis

Unlike Hp-MAP1 (NH_2-_AAGKVLKLLKKLL_-_COOH) and Hp-MAP2 (NH_2_-AAKKVLKLLKKLL-COOH) [32]. The sequence of the new analog, Hp-MAP3 (NH_2_-LLKKVLALLKKVL-COOH), displayed physicochemical features shared with AMPs, including net charge (+4), hydrophobicity (0.69), the content of hydrophobic residues (69%), and hydrophobic moment (0.73) features calculated on the *HeliQuest* website according to the Eisenberg scale. Further, the amino acids are distributed through a partially amphipathic α-helix, since the cationic face possesses a central Ala residue (hydrophobic) surrounded by 4 Lys. The hydrophobic face is composed of a combination of Leu and Val residues (Figure 1A). The tridimensional model of Hp-MAP3 displays the distribution of 4 Lys residues at N- and C-terminal (Figure 1B). The electrostatic surface shows that the addition of Ala between two Lys forms a small hydrophobic pocket, where electrostatic interactions with membranes might occur (Figure 1C). The synthetic peptide Hp-MAP3 was purified by reversed-phase high-performance liquid chromatography (HPLC-RF), with purity greater than 98%. The mass spectrum of Hp-MAP3 showed that the mass is 1.478 Da, corroborating the theoretical mass (Appendix A).

### 2.2. Evaluation of Antimicrobial Activity

The antibacterial activity of Hp-MAP3 and ciprofloxacin antibiotic expressed in MIC and MBC (Table 1). Among Gram-negative bacteria, *A. baumannii* was the most sensitive strain to the peptide, with an MBC of 2.7 μM, followed by *E. coli* and *K. pneumoniae*, both with 5.4 μM. *E. coli* KPC and *K. pneumoniae* KPC showed a MIC of 10.8 μM. Regards Gram-positive bacteria, *S. aureus* showed a MIC of 43.3 μM, suggesting a higher antimicrobial activity against Gram-negative bacteria. At (Appendix A) antibiogram assay against certain bacteria. (+) resistant at the concentrations tested.

### 2.3. Evaluation of Damage to Bacterial Membrane

To investigate whether the mechanism of action of Hp-MAP3 involves damage to the plasma membrane, the incubation of bacteria with DNA intercalant Sytox green was carried out (Figure 2). The Gram-negative *E. coli* KPC and Gram-positive *S. aureus* bacteria were incubated with Hp-MAP3 at MIC, at 21.6 and 43.3 μM, respectively. For both bacteria, an increase in Sytox fluorescence was noticed, suggesting that the peptide prompted damage to the plasma membrane. However, the time necessary to increase fluorescence varied according to the strain. For *E. coli* KPC, a linear increase in fluorescence was observed in 5 min, reaching a plateau of 100% from 20 min. For *S. aureus*, a peak of 100% of fluorescence was noticed within 5 min, followed by a decrease of 20% in fluorescence, maintained along the assay. These data confirm the occurrence of damage on the plasma membrane but evidence of differences in the interaction of Hp-MAP3 with Gram-negative and -positive membranes.

### 2.4. Evaluation of Antibiofilm Activity

As our best MIC result was against clinically isolated *A. baumannii*, it was used to determine the minimum inhibitory concentration biofilm (MICB) of Hp-MAP3. Thus, it was possible to observe that at 1 × MIC, the peptide inhibited 100% of biofilm formation, while at 0.5 × MIC it inhibited ~60% of the biofilm mass (Figure 3).

### 2.5. Evaluation of Pre-Formed Biofilm Activity

To evaluate the activity of Hp-MAP3 against the preformed biofilm of *A. baumannii*, the peptide was tested at different concentrations (Figure 4). The peptide showed potent activity at 173.2 μM, leading to ~100% death of the biofilm mass of *A. baumannii* cells. At the concentrations of 21.6 to 86.6 μM, the mass viability cells decreased by ~95 to ~90%.

### 2.6. Evaluation of Antifungal Activity

Hp-MAP3 was effective in inhibiting yeast growth, including the strains of *C. auris*, *C. albicans*, and *C. tropicalis*, with MIC ranging from 5.4 to 86.6 μM (Table 2).

### 2.7. Evaluation of Hp-MAP3 Cytotoxicity

The cytotoxic activity of the new Hp-MAP3 analog peptide against healthy MRC-5 cell lines and NCI-H292, RD, and HeLa tumor cells. The results showed that Hp-MAP3 inhibited 51% of the cellular viability of RD, 65% of the NCI-H292 line, and 85% of the HeLa line, all at the concentration of 21.6 μM (Figure 5). The value of IC_50_ was determined using the dose-response curve. Hp-MAP3 presented an IC_50_ for the tested lines MRC-5, RD, NCI-H292, and HeLa of 19.8, 24.0, 15.4, and 10.4 μM, respectively. The calculation of the selectivity index (IS) of the peptide for the tumor cell lines, RD, NCI-H292, and HeLa was 0.8, 1.2, and 1.8, respectively (Table 3).

### 2.8. Hemolytic Assay

We also evaluated the hemolytic activity of the Hp-MAP3 against murine erythrocytes. At low concentrations, up to 10.8 µM, a ~15% hemolysis was noticed. And at concentrations of 21.6 µM, the peptide had 25% hemolysis, and at concentrations of 43.3 and 86.6 µM, they exceeded the range of ~85% hemolysis (Figure 6). These data suggest that the rate of hemolysis might be modulated according to the concentration of peptide in the solution.

### 2.9. Secondary Structure of Hp-MAP3 in Different Environments

Structural features of Hp-MAP3, regards its secondary structure, were obtained through circular dichroism (CD) studies, incubating the peptide in solutions that mimic different environments (Figure 7). In an aqueous solution, Hp-MAP3 showed a *random coil* structure, characterized by a negative band at 195–200 nm and an absence of signal ~220 nm. The pattern of the signal was altered when Hp-MAP3was incubated with 50% trifluoroethanol (TFE) and SDS 30 mM micelles. The CD signals showed a positive band at approximately 192 nm and negative bands at 208 and 222 nm. The changes in the CD signal and the presence of negative bands at 208 and 222 nm indicate a conformational transition that indicates the acquisition of α-helical content, suggesting that Hp-MAP3 possesses a secondary structure modulated according to the environment [38].

## 3. Discussion

The physical-chemical properties of AMPs, such as charge, length, hydrophobicity, and secondary structure, are interrelated. Modifications in these parameters result in significant changes in both physical and biological features. Understanding and controlling these interrelationships is essential to the conception of new peptides with greater potential and specificity [39].

AMPs emerged as potential candidates for the treatment of microbial infections [40]. Therefore, investigating their activity in conditions similar to clinical application, in addition to antimicrobial and hemolytic trials, is crucial for greater knowledge about the structure, design, and activity. We aimed to design an AMP inspired by the temporin-PTa sequence, performing amino acid substitutions that preserve both secondary structure (α-helix) and amphipathicity. The combination of cationicity and hydrophobicity are determinants for the activity of an AMP by biological membranes. The occurrence of hydrophobic amino acid residues (>50%) contributes to the insertion of AMPs in the phospholipid membrane bilayer core [40,41]. For antibacterial activity, AMPs must establish numerous electrostatic interactions with the lipoteichoic acid (LTA) and lipopolysaccharide (LPS), present in Gram-positive and -negative bacteria cell walls, respectively [42].

Mishra and Wang reported the design of four temporin-PTa analogs, named DFTamP1, DFTamP1-p, DFTamP1-pi, DFTamP1-pv [43]. The temporin-PTa analogs exhibited MIC from 25 to >120 μM. Previously studies from Souza e Silva and collaborators demonstrated that temporin-PTa present MIC values of 20 to 45 µM and the two analogs Hp-MAP1 and Hp-MAP2 were actives also against bacteria Gram-negative and -positive with MIC values of 2 to 45 µM [32]. Our analog, Hp-MAP3, showed MIC ranging from 2.7 to 43.3 μM, demonstrating a similar antimicrobial activity. In this same study, the authors designed two analogs to investigate the contribution of Leu residues in the primary sequence of a peptide, and they showed improvement in bacterial activities with a MIC of 12.5 μM for *E. coli* [43]. The results of Hp-MAP3 that has six leucine residues in their primary sequence. Hp-MAP3 shared MIC and MBC similar to those observed in other AMPs, such as magainin-2, buforine-2, and melittin [44], beyond six temporin analogs, temporin-CPb, 1Ga, 1OLa, 1Spa, 1Oc, and CPa [45].

The bactericidal effect of Hp-MAP3 involves damage to plasma membranes, a mechanism shared among most amphipathic α-helical AMPs [46,47]. *Sytox green* is a high-affinity nucleic acid intercalant that does not cross the intact membranes of living cells but easily penetrates cells with damaged plasma membranes [48]. The time required for Hp-MAP3 to play its antibacterial role corroborates the finding from AMPs from the temporin family [49,50].

The intrinsic biofilm resistance to antibiotic therapy has made biofilm-related infections a relevant clinical problem [51], besides being significantly higher than those of bacteria in the plankton state [52]. Infections caused by biofilms are more difficult to control and can lead to death [53,54]. Bacterial species such as *S. aureus*, *P. aeruginosa*, *K. pneumoniae*, and *A. baumannii* are common in infections associated with biofilms, including infection in sputum, pulmonary infections, patients with cystic fibrosis, related to implants, such as venous catheters, endotracheal tubes, prostheses [51]. Grassi and collaborators reported a temporin-1TB analog peptide, named TB_KKG6A, that also displayed the ability to inhibit bacterial biofilm formation [55]. In previous studies, the results showed that synthetic temporin-PTa did not inhibit *E. coli* KpC and *A. baumannii* biofilms formation. In contrast, the analogs Hp-MAP1 and Hp-MAP2 presented MICB at 23 and 43 μM, respectively for *E. coli* KpC. In addition, the same analogs were actives for *A. baumanni* with MICB values of 92 and 87 μM, respectively [32]. Therefore, in this study, Hp-MAP3 exhibited antibiofilm properties at a concentration of 2.7 μM against *A. baumanni* improving ~35 × MIC in comparison with the Hp-MAP1 and Hp-MAP2 analogs [32]. The inhibition of biofilm by Hp-MAP3 was noticed at MIC, supporting that the antibiofilm activity occurred by direct death of biofilm-forming bacteria still in their planktonic stage instead of modulating pathways related to biofilm formation [9,56]. For the eradication of the biofilm already formed, it takes a larger amount of peptide [57]. This is shown in studies with different temporins where all peptides tested showed a potent activity at 50 and 100 μM leading to ~100% death of biofilm cells of *S. aureus* ATCC 25923 [58]. In the eradication of the *A. baumannii* biofilm, the Cec4 peptide eradicated the biofilm at a concentration of 64 × MIC [59]. The same characteristic of Hp-MAP3 eradicated the biofilm at a concentration of 64 × MIC for the same species of bacteria.

The antifungal properties of Hp-MAP3 were confirmed against different *Candida* strains. The design of Hp-MAP3 prompted a higher efficiency when compared with other AMPs with antifungal activity, such as the synthetic EcDBS1R6 (100 μM) [58] and the natural EC1-17, from *Echinus esculentus* (>122.8 μM) [59]. The incidence of fungal infections is overwhelming in pharmaceutical, medical, and basic science [60,61]. Only a limited number of licensed antifungals are available for combating fungal infections [62]. The membranes of lower eukaryote, such as yeasts, is known for the presence of ergosterol. The relationship between AMP activity and toxicity was explained by surface charge in cell membranes, and the role of other lipids, including sterols, was secondary [63]. Cholesterol and ergosterol share a similar structure, the increase of sterol in membranes enables a less hydrophobic environment, reducing interactions between AMPs and membranes, and increasing the exclusion of AMPs from the nucleus of the hydrophobic membrane [63]. For this reason, the activity of AMPs against bacteria, protozoa, yeasts, and other fungi varies [63,64]. The activity of AMPs against membranes of higher organisms, such as mammals also varies. Hp-MAP3 was able to overcome the ordering effect from ergosterol in yeast membranes, presenting activity against different *Candida* strains. However, the incidence of hemolysis against murine erythrocytes and the effect on MRC5 cell viability suggest that further pharmaceutical presentations should be planned for the clinical application of AMPs.

AMPs have shown promising results as new therapeutic agents in cancer, as they are membrane selective [65]. Brevinin-2R is an antimicrobial peptide from the cutaneous secretion of *Rana ridibunda* that has a membrane-disruptive mode of action. The results demonstrated that brevinine inhibited the proliferation of 50% of HeLa cells at ~80 μM [66]. Another example is aurein peptides 1.2 and 3.1 from *Litoria* sp., which showed activity against the NCI cancer cell line, with LC_50_ values of 4 and 10 μM, respectively [67]. Furthermore, data on RD cells and peptides are scarce in the literature nonetheless, recent studies have shown that phospholipase (BmPLA2) from *Bothrops moojeni* demonstrated an IC_50_ of ~0.14 μM [68].

Abramo and collaborators determined by circular dichroism spectroscopy (DC) the conformation of two synthetic temporins, temporin-L and -A, recording spectra in water and TFE. For both peptides, the experiments showed that TFE favored a gradual transition from a random coil to an α-helical structure [69]. In previous studies, the amphipathic peptides, Hp-MAP1 and Hp-MAP2 showed in silico helical stability in the environments with TFE and SDS [32]. The bioassay results of Hp-MAP3 demonstrated a random coil structure in an aqueous solution and α-helix structure in the presence of TFE and SDS micelles. There is an agreement that an α-helical AMPs that present sequences with hydrophobic amino acid residues contribute to a disordered structure in an aqueous environment [70].

Based on its amphiphilic properties, after an initial electrostatic interaction with membranes, AMP presents a secondary structure driven by the formation of several interactions [20,71] once interacted with the membrane, AMPs trigger one of several modes of lysis [72]. Several studies indicate that temporins can enter and damage the cell membrane as part of their mechanism of action [49,73,74,75]. Divided into two stages: first, the peptides by electrostatic interactions, accumulate in the membrane of bacteria with groups negatively charged on the surface, then the hydrophobic parts of the AMPs are in the lipid bilayer, causing direct ruptures in the membranes, which can be forming pores, toroidal pore, disordered toroidal pore or carpet mechanisms. This results in membrane depolarization, loss of specific composition, and leakage of essential metabolites [76]. Therefore, the hydrophobicity together with the proper hydrophobic and net charge moment, raises the chances of the peptide assuming an α-helical conformation, improving the interaction with membranes and consequently improving its activity [77].

## 4. Conclusions

The Hp-MAP3 peptide was rationally designed using the temporin-PTa primary sequence as a template. In addition, adjustments were made so that the physicochemical characteristics, such as cationic charge and hydrophobicity content, were optimized for membrane permeabilization. Although the Hp-MAP3 showed moderate hemolytic activity, it is able to eradicate pathogenic bacteria cells within minutes and at low concentrations, exhibiting activity against different fungal strains and cytotoxicity against cancer cells. Therefore, Hp-MAP3 is a promising candidate against pathogenic microorganisms, providing important information to assist in the design of new peptide sequences.

## 5. Material and Methods

### 5.1. Rational Design and Synthesis

The rational design strategy was to use the primary sequence of temporin-PTa [78] in the obtainment of *Hylarana picturata-Multi Active Peptide* analog (Hp-MAP3). A similar rational strategy proposed by Souza e Silva and collaborators [32] was applied. The replacements of Ser^4^ and Pro^10^ in the temporin-PTa sequence by Lys represent an alteration of uncharged polar and an α-helix distribution residue, respectively. Substitution of Lys by an Ala aid in the extension of the α-helix segment. Finally, the Ile and Phe residues were replaced by Leu and Val to reduce the hydrophobicity of the peptide. The helix wheel projections and the physicochemical properties, including charge, hydrophobicity, and hydrophobic moment were obtained using the *HeliQuest* server [79]. The theoretical three-dimensional models were built by comparing analogs structures present in databases, through the *I-TASSER* server. With the PDB model, it was possible to visualize it using the *PyMOL* version 2.3 program, thus extracting the three-dimensional representations of the peptides. The peptides were synthesized by a solid-phase method using the N-9-fluorenylmethoxycarbonyl (Fmoc), purified by reversed-phase high-performance liquid chromatography (RP-HPLC) with >95% purity on an (acetonitrile: H2O:TFA) gradient and confirmed by ion spray mass spectrometry by Aminotech Company (São Paulo, Brazil).

### 5.2. Minimal Inhibitory Concentration (MIC) and Minimal Bactericidal Concentration (MBC)

The determination of MIC was performed against *Escherichia coli* (clinical isolate LACEM, Brazil), *Escherichia coli* (KpC), *Klebsiella pneumoniae* (ATCC), *Klebsiella pneumoniae* (KPC + 39), *Acinetobacter baumannii* (clinical isolate the LACEM, Brasília, Brazil), *Pseudomonas aeruginosa* (ATCC 40), *Staphylococcus aureus* (clinical isolate LACEM, Brasília, Brazil). The bacteria were plated in Mueller Hinton agar (MHA) and incubated at 37 °C overnight. Three isolated colonies were inoculated in 5 mL of Mueller Hinton broth (MHB) and incubated at 200 rpm at 37 °C overnight. Bacterial growth was monitored by a spectrophotometer at 600 nm. The MIC was performed by the microdilution method, using a 96-wells microplate, with a final bacterial concentration of 5 × 10^5^ CFU·mL^−1^. Hp-MAP3 was assayed at concentrations ranging from 1.3 to 43.3 μM. Bacterial suspension in MHB was used as a negative control. The microplates were incubated at 37 °C for 18 h, and the reading was performed on a Multiskan Go microplate reader (Thermo Scientific: Waltham, MA, USA) at 600 nm. The MIC was determined as the lowest peptide concentration in which there was no bacterial growth. The evaluation of the minimum bactericidal concentration (MBC) was dependent on the results of the MIC. Samples of 10 μL of each well were removed, plated in MHA, and incubated at 37 °C for 24 h. The CBM was determined as the lowest peptide concentration in which no bacterial growth was detected. The determinations were performed in three independent assays, carried out in triplicate [80].

### 5.3. Membrane Permeabilization Using Sytox Green

The bacterial membrane permeability was investigated using the Sytox green dye, as described by Mohanran and collaborators, with modifications proposed by Almeida and collaborators [40,81]. In the assay, *E. coli* (KPC + 34) and *S. aureus* (clinical isolate) were cultivated in MHB broth for 18 h to 37 °C, prepared at an optical density of 600 nn of 0.5, in sodium phosphate buffer 10 mM, pH 7.0. Then, 280 μL of bacterial suspension was transferred to 96-well black microplates, where 10 μL of Sytox green (30 μM) were added, and incubated for 10 min to 37 °C. Subsequently, 10 μL of Hp-MAP3, at a concentration 30 times higher than MIC, was added to each well, and the kinetic assay was performed for 50 min, with readings every 5 min. The assay was performed with fluorescence reading, excitation at 485 nm, and emission at 520 nm, in a Varioskan Lux microplate reader (Thermo Scientific: Waltham, MA, USA). The negative control of membrane damage was performed with bacteria incubated with 10 μL of sodium phosphate buffer 10 mM, pH 7.0. Three independent experiments were carried out in triplicate.

### 5.4. Minimum Inhibitory Concentration of Biofilm (MICB)

The biofilm formation was obtained using Basal Medium 2 (BM2) (potassium phosphate 62 mM, (NH_4_)_2_ SO_4_ 7 mM, MgSO_4_ 2 mM, FeSO_4_ 10 mM, and 0.4% glucose). Bacterial cultures of *Acinetobacter baumannii* (clinical isolate) were cultivated for 18 h in MHB and diluted (1:100, v:v) in BM2. The bacterial suspension was plated in 96 round bottom well plates containing Hp-MAP3 in serial dilutions, from 1.3 to 43.3 μM (final volume of 100 μL). The microplates were incubated for 24 h at 37 °C. Negative growth control contained only bacterial cells. To evaluate the formation of biofilm, the medium was removed from the microplates, and the wells were washed twice with deionized water. The adherent cells were flushed with 0.01% crystal violet for 20 min, after the wells were washed twice with deionized water, air-dried, and the violet crystal adhered to the cells was solubilized with 110 μL of 60% ethanol. Biofilm formation was measured at 595 nm. All absorbance readings were performed with the microplate reader Multiskan Go (Thermo Scientific). Three independent experiments were carried out in triplicate.

### 5.5. Antibiofilm Activity

A preformed biofilm of *A. baumannii* was obtained as previously reported, with some modifications [58]. The microbial culture was grown at 37 °C to an optical density (OD) of 1.6 (λ = 600 nm) and then diluted to a cell density of 1 × 106 colony-forming units CFU·mL^−1^. Aliquots of 100 µL were dispensed into the wells of a 96-multiwell plate, which was incubated for 20 h at 37 °C for biofilm formation. After the incubation time, the medium containing planktonic cells was aspirated from the wells and replaced by 100 µL of phosphate-buffered saline (PBS) to remove any non-adherent cells. A PBS wash was performed twice. After washing, each well was filled with PBS supplemented with different two-fold serial dilutions of Hp-MAP3(from 1.3 to 173.2 µM), and the plate was then incubated for 2 h at 37 °C. After peptide treatment, the wells were rinsed twice with PBS, as indicated above, and 20 µL of MTT (0.5 mg·mL^−1^) was dispensed in each well in order to evaluate biofilm cell viability. A higher intensity of purple color corresponds to a higher percentage of metabolically-active cells and consequently to higher cell viability. The plate was incubated and protected by light at 37 °C for 4 h, and the reaction was stopped by adding the solubilized solution. The absorbance of each well was recorded at 570 nm using a microplate reader Multiskan Go (Thermo Scientific, Waltham, MA, UnSA), and the percentage of biofilm viability was calculated with respect to the untreated samples.

### 5.6. Yeast Strains and MIC Determinations

*Candida albicans* strains ATCC 90029, *Candida tropicalis* ATCC 750, *Candida krusei* ATCC 6258, *Candida parapsilosis* ATCC 22019, *Candida glabrata* ATCC 9030, *Candida auris* CBS 10913 e four *Candida auris* (2015/466, 2015/467, 2015/468, 2015/470) clinical isolates from Venezuelan Outbreak [82] were used. The strains were stored at −80 °C in nutrient broth supplemented with 2% glycerol. Before experimentation, each strain was streaked for single colonies on Sabouraud Dextrose (SDA) agar plates and incubated overnight at 37 °C. The antifungal activity of the designed peptides was determined according to a standardized broth microdilution method (Clinical and Laboratory Standards Institute (CLSI) document M27-A2). Briefly, the yeast colonies from 24 h old cultures of *Candida* species were picked and resuspended in 5 mL of sterile 0.145 mol.L^−1^ saline and adjusted to cells density of 1 × 10^6^ to 5 × 10^6^ cells·mL^−1^. The yeast stock suspension was then diluted to obtain a starting inoculum of 5.0 × 10^2^ to 2.5 × 10^3^ cells·mL^−1^. The peptide was dissolved in water and prepared to a stock concentration of 2 mg.L^−1^, then a serial dilution in RPMI 1640 growth medium buffered with 3-morpholinopropane-1-sulfonic acid (MOPS) was prepared, in a volume of 30 µL per well, giving final concentrations ranging from 43.3 to 1.3 μM in sterile bottomed 96-well microplates. A volume of 30 µL of standardized yeast suspension was added to each well. Plates were incubated for 48 h at 37 °C and monitored by 530 nm. The MIC was defined as the lowest concentration that inhibited 90% growth of *Candida* species.

### 5.7. Cell Viability Test

In order to evaluate the cytotoxicity of Hp-MAP3, the viability in vitro of tumor cells NCI-H292 (ATCC CRL-1848), RD (ATCC CCL-136), HeLa (ATCC CCL-2), and healthy MRC-5 (ATCC CCL-171), purchased from the Cell culture center of the Adolf Lutz Institute (São Paulo, Brazil), were performed as the follows [83]. The cells were plated to 1 × 10^4^ cells.well^−1^ in 96-well microplates and treated with 100 μL of different Hp-MAP3 concentrations (from 21.6 to 0.33 μM) for 24 h. As a negative control, the cells were incubated with a culture medium. After the incubation period, the supernatant was removed and 100 μL of enzymatic reduction of 3-bromide(4,5-demethyliazole-2-il)-2,5-diphenyltetrazozole (MTT, Sigma-Aldrich; St Louis, MO, USA) solution (1 mg·mL^−1^ diluted in culture medium) were added. After 4 h of incubation, the formazan crystals were resuspended with 100 μL dimethylsulfoxide (DMSO) and read at 570 nm in the MultiSkan Go microplate reader. Three independent experiments were carried out in triplicate. The cell viability was calculated from the following formula.

### 5.8. Hemolytic Assay

The assay was developed according to Kim and collaborators, with modifications [84]. The erythrocytes of Swiss *Mus musculus* mice were washed three times with phosphate buffer 50 mM (PBS), pH 7.4. A serial dilution of Hp-MAP3 (1.3 to 43.4 µM) was incubated with 1% erythrocyte suspension. The samples were incubated at room temperature for 60 min. The release of hemoglobin was monitored by measuring the absorbance of the supernatant at 415 nm. The negative control was determined by incubating erythrocytes in the presence of PBS 50 mM, pH 7.4, while a solution of 1% Triton X-100 was used as a positive control. This experiment was approved by the CEUA of the Universidade Católica Dom Bosco under the number 014/2018. The hemolytic activity was performed in triplicate.

### 5.9. Circular Dichroism Spectroscopy (CD)

The CD analyses were performed on a Jasco J-1100 spectropolarimeter (Jasco Inc., Tokyo, Japan) using a quartz bucket with 1 mm of the optical path. The spectrum from 260 to 185 nm was collected with steps at resolution from 0.1 nm to 100 nm.s^−1^, at 25 °C, with an average of 5 accumulated scans for each condition. A stock solution of 120 μM Hp-MAP3 was prepared in water and used in a working solution of 30 μM. The secondary structure of the peptide was analyzed in the presence of water, 50% trifluoroethanol (TFE), and 30 mM SDS. The data were converted to molar ellipticity (θ), according to the equation:(1)$$\left[\theta \right]=\frac{\theta }{10*C*l*\mathrm{nr}}$$
where θ is the ellipticity measured in milliseconds, C is the concentration of the peptide (M), l is the length of the bucket path and n_r_ is the number of amino acid residues. The fractional content of α-helix, h_H_, was estimated by the equation:(2)$$fH=\frac{\theta 222-\;\theta C}{\theta H-\;\theta C}$$
where θC = 2220-53, θH = (250 T−44,000) (1–3/*n*), where T is the temperature in Celsius, and n is the number of amino acid residues of the peptide. The values θC and θH represents, respectively, the mean ellipticity limit values at 222 nm (θ222) for disordered and α-helix conformation.

### 5.10. Statistical Analysis

The statistical significance of the experimental results was determined by one-way Student’s *t*-test or one-way analysis of variance (ANOVA) followed by Dunnett’s test. Values of *p* < 0.05 were considered statistically significant. *GraphPad Prism* version 8.0 was used for all statistical analyses.

## Figures and Tables

**Figure 1 toxins-15-00042-f001:**
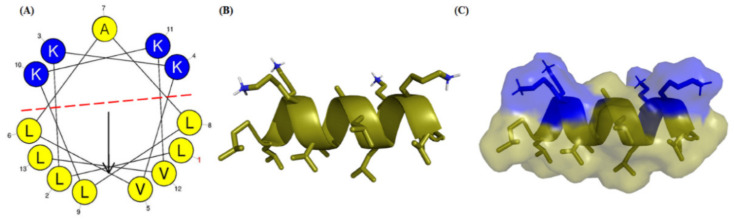
The α-helical projection and 3D structure of the Hp-MAP3 peptide sequence. (**A**) Distribution of prediction amino acid residues by *HeliQuest*. (**B**) Theoretical three-dimensional structure of hp-MAP3 provided by *I-TASSER* with lateral chain arrangement. (**C**) Theoretical three-dimensional structure of the Hp-MAP3 with the electrostatic surface visualized by the *PyMol* program. The positively charged hydrophilic residues is represented in blue and the hydrophobic aliphatic are represented in yellow.

**Figure 2 toxins-15-00042-f002:**
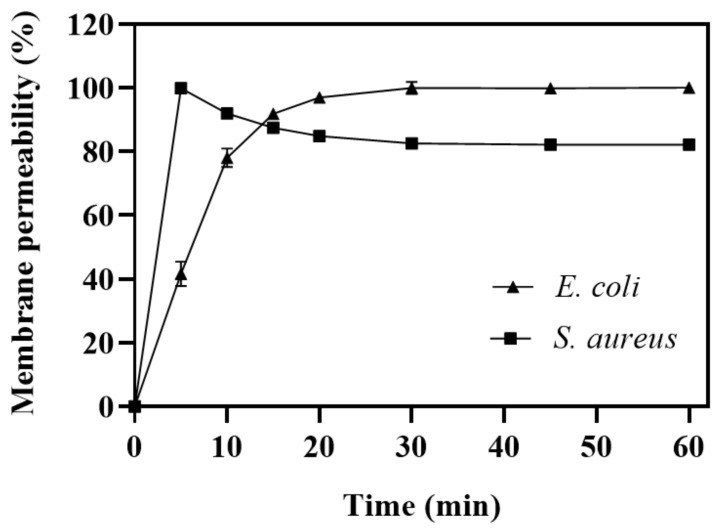
Evaluation of the level of damage to the bacterial membrane. Uptake of Sytox green by *E. coli* (▲) and S. aureus (■) treated with Hp-MAP3 at concentrations of 10.83 and 43.3 μM, respectively. The Sytox Green uptake was measured during 60 min of incubation with Hp-MAP3.

**Figure 3 toxins-15-00042-f003:**
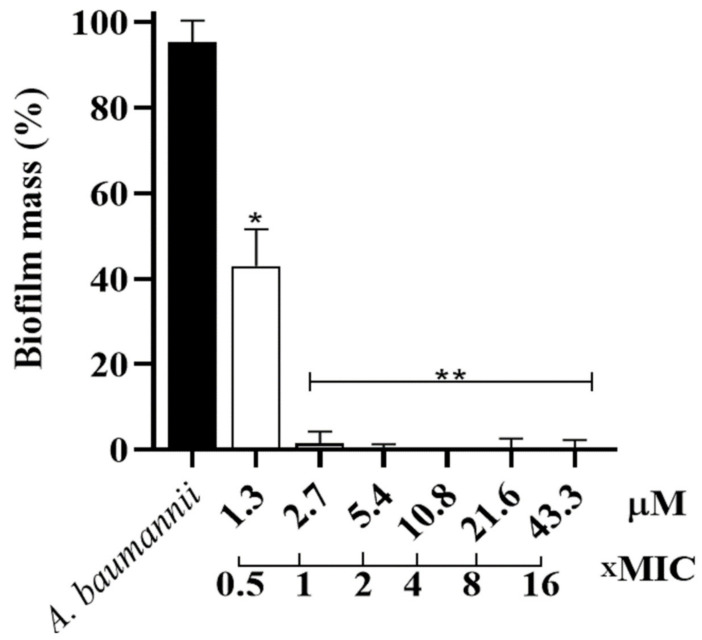
Biofilm mass as an inhibition percentage against the *A. baumannii* strain at different concentrations and the concentration of xMIC. ANOVA followed by Dunnett’s test (**) *p* ˂ 0.01 and (*) *p* ˂ 0.05.

**Figure 4 toxins-15-00042-f004:**
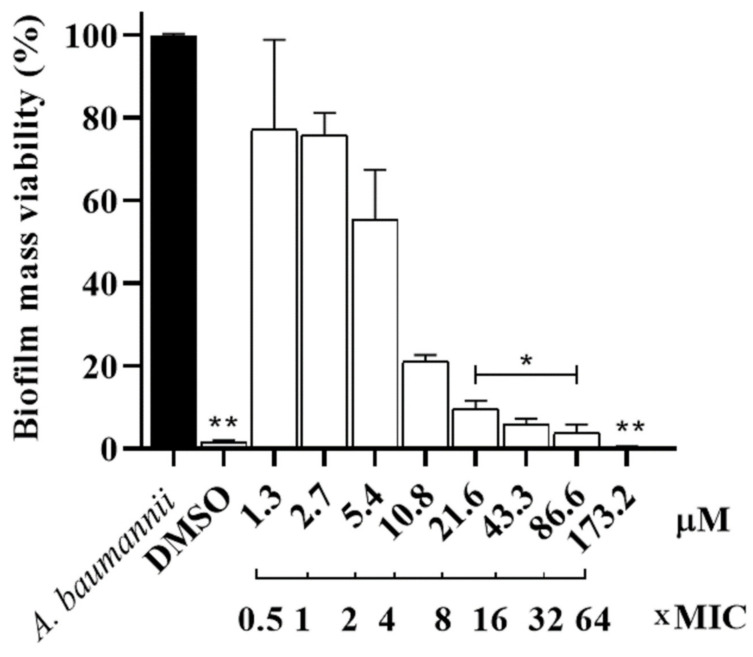
Biofilm mass viability as an inhibition percentage against the *A. baumannii* strain at different concentrations and the xMIC concentration. ANOVA followed by Dunnett’s test (**) *p* ˂ 0.01 and (*) *p* ˂ 0.05.

**Figure 5 toxins-15-00042-f005:**
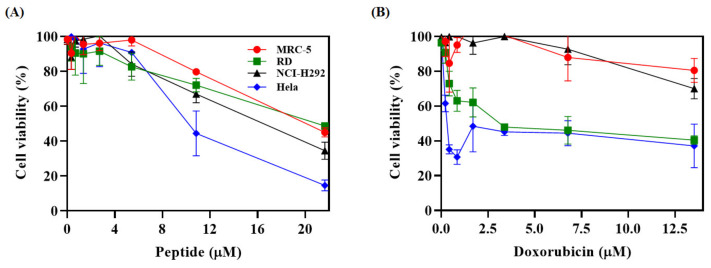
Evaluation of the cytotoxicity of Hp-MAP3 and commercial anticancer drug doxorubicin against healthy cell MRC-5 and RD, NCI-H292 and HeLa tumor cells. (**A**) Hp-MAP3 peptide and (**B**) antibiotic.

**Figure 6 toxins-15-00042-f006:**
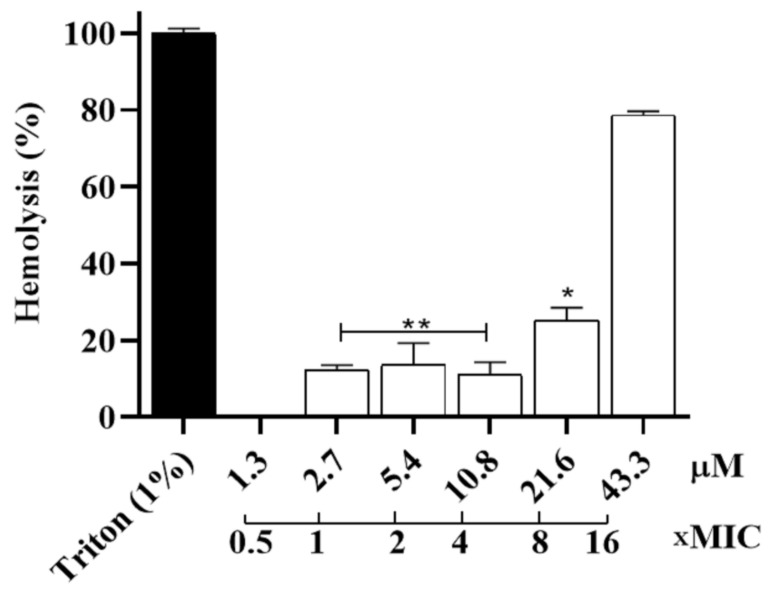
Evaluation of the hemolytic peptide Hp-MAP3 against murine erythrocytes and the concentration of xMIC. ANOVA followed by Dunnett’s test (**) *p* ˂ 0.01 and (*) *p* ˂ 0.05.

**Figure 7 toxins-15-00042-f007:**
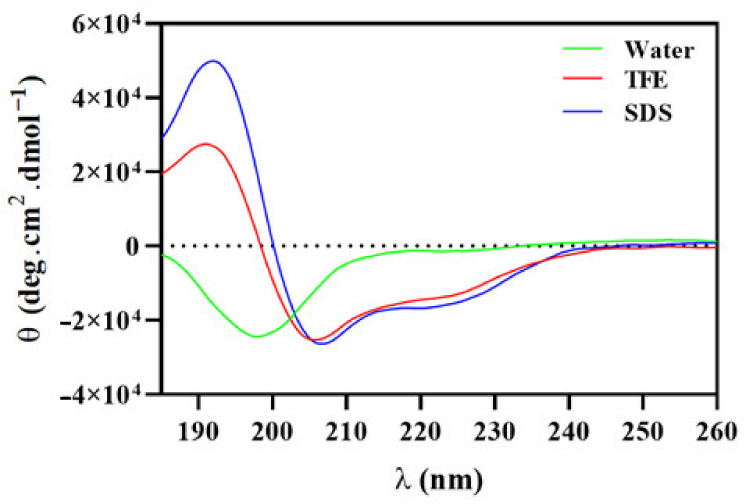
Conformational changes of Hp-MAP3 were evaluated by CD in water, TFE and SDS environments. Hp-MAP3 spectra in water (green line), 50% TFE (red line) and 30mM SDS (blue line).

**Table 1 toxins-15-00042-t001:** Minimal inhibitory concentration and minimal bactericidal concentration of the peptide Hp-MAP3 against pathogenic bacterium.

Microorganism	Hp-MAP3 (µM)		Ciprofloxacin (µM)	
Gram-Negative	MIC ^a^	MBC ^b^	MIC ^a^	MBC ^b^
*Acinetobacter baumannii* (clinical isolated)	2.7	2.7	54.4	>100.8
*Escherichia coli* (clinical isolated)	5.4	21.6	54.4	>100.8
*Escherichia coli* (KPC + 34)	10.8	21.6	100.8	100.8
*Klebsiella pneumoniae* (ATCC 36)	5.4	5.4	100.8	100.8
*Klebsiella pneumoniae* (KPC + 39)	10.8	21.6	54.4	54.4
*Pseudomonas aeruginosa* (ATCC)	43.3	43.3	6.3	6.3
Gram-positive				
*Staphylococcus aureus* (clinical isolated)	43.3	43.3	6.3	6.3

**^a^** Minimal inhibitory concentration; ^b^ Minimal bactericidal concentration; Values are expressed from experiments performed with technical and biological replicates in triplicate.

**Table 2 toxins-15-00042-t002:** The Minimal inhibitory concentration of the peptide Hp-MAP3 against fungal strains.

Microorganism	Hp-MAP3 (µM)
*Candida albicans* ATCC 90029	21.6
*C. auris* CBS 10913	10.8
*C. auris* 2015/466	5.4
*C. auris* 2015/467	10.8
*C. auris* 2015/468	10.8
*C. auris* 2015/470	10.8
*C. glabrata* ATCC 9030	86.6
*C. krusei* ATCC 6258	43.3
*C. parapsilosis* ATCC 22019	43.3
*C. tropicalis* ATCC 750	21.6

**Table 3 toxins-15-00042-t003:** IC_50_ values and Hp-MAP3 peptide selectivity index against healthy MRC-5 and tumor cell lines RD, NCI-H292 and HeLa.

Cell Line	Hp-MAP3	
	IC_50_ *	IS **
MRC-5	19.83	-
RD	24.01	0.82
NCI-H292	15.44	1.28
HeLa	10.46	1.89

* IC_50_ corresponds to a minimum concentration effective to inhibit 50% of cell growth; ** IS corresponds to the peptide selectivity index comparing the activity in the tumor cell in relation to the healthy cell.

## Data Availability

Not applicable.

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
