# Peer review of "Evaluation of the Synthetic Multifunctional Peptide Hp-MAP3 Derivative of Temporin-PTa"

_toxins, 2023, doi:10.3390/toxins15010042_

Round 1
Reviewer 1 Report
Dear editors
This is an analysis paper of a synthetic analogue of an antimicrobial peptide, it was tested in several microorganisms and eukaryotic cell lines and erythrocytes.
After reading and analyzing the paper, I conclude that it is well written.
The introduction is cohesive, materials and methods suited to antimicrobial analysis papers.
The results were well demonstrated and the discussion well developed.
Please correct figure 2, replace "TEMPO" with "TIME", on the X axis
Best regards
Author Response
Review 1:
1) This is an analysis paper of a synthetic analogue of an antimicrobial peptide, it was tested in several microorganisms and eukaryotic cell lines and erythrocytes. After reading and analyzing the paper, I conclude that it is well written. The introduction is cohesive, materials and methods suited to antimicrobial analysis papers. The results were well demonstrated and the discussion well developed.
Response: The authors appreciate all the time dedicated to improving the work.
2) Please correct figure 2, replace "TEMPO" with "TIME", on the X axis
Response: The authors agree and the X axis was modified as suggested.

Reviewer 2 Report
The article is deal with the structural and antimicrobial activities of synthetic antimicrobial peptide, derivative of natural temporins. The topic discussed is very important for the treatment and prevention of infectious diseases.
I would like to make a few comments:
1) Perhaps, in the title of the article the words “inspired by” replace on “analog of” or “derivative of”…
2) line 5 and 62-63: AMPs have also anti-viral, anti-parasitic, and immunomodulatory activities.
Cite, please, the recent publications about these activities (2022-2021), for example from MDPI journals.
3) It is necessary to add to the keywords word “temporins”
4) line 24: “Several infectious diseases have become treatable”
Perhaps, it is better to use another phrase, for example “A large number of infectious diseases have become treatable” or “Many of infectious diseases..” or something like that
5) line 86: It will be better to indicate the amino acid sequences for analogs Hp-MAP1 and Hp-MAP2 in comparison with amino acid sequence for Hp-MAP3.
6) There is a mistake on the figure 5: the scale of the axis “Hemolysis in (%)” should be changed; on the place of 120 should be 100, etc .
In the text it is written: “10.8 µM, a ~15% of hemolysis was noticed” but according to the figure 5 it is 10.8 µM, a ~25% of hemolysis was noticed
7) Figure 5: “Hemolysis in (%)” better to change on “Hemolysis, %”
8) Figure 5: the words “[Peptide] (µM)” better to change on “Peptide concentration, µM”
9) line 174: when you use for the first time “CD”, please wright full words “circular dichroism (CD)”
10) Figure 6: when you title the axes use comer, not brackets : “λ, nm”, the same to another axis for molar ellipticity
11) In Discussion it will be better to add the comparison of the activities and stability of temporins Hp-MAP1, Hp-MAP2 with Hp-MAP3
12) In Material and Methods add the item Statistical Analysis
Author Response
Review 2:
- Perhaps, in the title of the article the words “inspired by” replace on “analog of” or “derivative of”…
Response: The authors agree and in the title of the article the words “inspired by” was replaced on “derivative of”.
Now reads:
(Title: Page 1, line 1):
“Evaluation of the synthetic multifunctional peptide Hp-MAP3 derivative of temporin-PTa”
2) Line 5 and 62-63: AMPs have also anti-viral, anti-parasitic, and immunomodulatory activities.
Response: The authors agree and as suggested the information about activities were added in the abstract and introduction.
Now reads:
(Abstract: Page 1, line 5):
“In recent years, antimicrobial peptides isolated from amphibian toxins have gained attention as new multifunctional drugs interacting with different molecular targets”.
(Introduction: Page 2, line 71):
“In recent years, AMPs have gained increasing attention as new antimicrobial candidates to combat infections caused by bacteria, forming biofilm bacteria, fungi, virus, protozoa, tumoral cells in culture and immunomodulators [22–26]”.
3) It is necessary to add to the keywords word “temporins”:
Response: The authors agree and the word “temporins” was added to the keywords.
4) line 24: “Several infectious diseases have become treatable” Perhaps, it is better to use another phrase, for example “A large number of infectious diseases have become treatable” or “Many of infectious diseases..” or something like that:
Response: The authors agree and the expression “A large number of infectious diseases have become treatable”, was added as suggested.
Now reads:
(Introduction: Page 1, line 25):
“A large number of infectious diseases have become treatable”
5) line 86: It will be better to indicate the amino acid sequences for analogs Hp-MAP1 and Hp-MAP2 in comparison with amino acid sequence for Hp-MAP3.
Response: The authors agree. Good reviewer placement, the addition was done as suggested.
Now reads:
(Results: Page 3, line 98):
“Unlike Hp-MAP1 (NH2-AAGKVLKLLKKLL-COOH) and Hp-MAP2 (NH2-AAKKVLKLLKKLL-COOH) [39]”.
6) There is a mistake on the figure 5: the scale of the axis “Hemolysis in (%)” should be changed; on the place of 120 should be 100, etc.
Response: The authors agree and. The axis “Hemolysis in (%)” was changed to 100%.
- In the text it is written: “10.8 µM, a ~15% of hemolysis was noticed” but according to the figure 5 it is 10.8 µM, a ~25% of hemolysis was noticed
Response: The authors does not agree. The results reality for 10.8 µM represent ~15% of hemolysis as is written in the text and Figure 5.
7) Figure 5: “Hemolysis in (%)” better to change on “Hemolysis, %”, Figure 5: the words “[Peptide] (µM)” better to change on “Peptide concentration, µM”
Response: The authors agree and the modification were realized in the Figure 5.
8) line 174: when you use for the first time “CD”, please wright full words “circular dichroism (CD)”
Response: The authors agree and the modification was performed.
9) Figure 6: when you title the axes use comer, not brackets: “λ, nm”, the same to another axis for molar ellipticity.
Response: The authors agree and the modification were realized in the Figure 6.
10) In Discussion it will be better to add the comparison of the activities and stability of temporins Hp-MAP1, Hp-MAP2 with Hp-MAP3.
Response: The authors agree and the discussions was modified as suggested.
Now reads:
(Discussion: Page 8, line 230):
“Previously studies from Souza e Silva and collaborators demonstrated that temporin-PTa present MIC values of 20 to 45 µM and the two analogs Hp-MAP1 and Hp-MAP2 were actives also against bacteria Gram-negative and -positive with MIC values of 2 to 45 µM [39].”
(Discussion: Page 9, line 255):
“In previous works we showed that synthetic temporin-PTa did not inhibit E. coli KpC and A. baumannii biofilms formation. In contrast, the analogs Hp-MAP1 and Hp-MAP2 presented MICB at 23 and 43 μM, respectively for E. coli KpC. In addition, the same analogs were actives for A. baumanni with MICB values of 92 and 87 μM, respectively [39]. Therefore, in this study, Hp-MAP3 exhibited antibiofilm properties at concentration 2.7 μM against A. baumanni improving ~35 ×MIC in comparison with the Hp-MAP1 and Hp-MAP2 analogs.
(Discussion: Page 10, line 296):
“In previous works the amphipathic peptides Hp-MAP1 and Hp-MAP2 showed in silico helical stability in the environments with TFE and SDS [39]”.
11) In Material and Methods add the item Statistical Analysis.
Response: The authors agree and the section Statistical Analysis was added.
Now reads:
(Material and Methods: Page 13, line 453):
“5.9 Statistical Analysis
The statistical significance of the experimental results was determined by one-way Student’s t-test or one-way analysis of variance (ANOVA) followed by Dunnett test. Values of P< 0.05 were considered statistically significant. GraphPad Prism version 8.0 was used for all statistical analyses”.

Reviewer 3 Report
This manuscript reports the design, synthesis and biological evaluation of a peptide, Hp-MAP3. The authors initially identified temporin A-PTa as the starting point of the study, followed by designing and synthesising the new analogue, Hp-MAP3, by modifying the amino acids in temporin A-PTa sequence. MIC testing against various bacterial and fungal strains suggested that Hp-MAP3 possesses moderate to high antimicrobial activity. Mechanistic study suggested that Hp-MAP3 mimics the mechanism of AMPs. The cytotoxicity of Hp-MAP3 was also confirmed by MTT and hemolytic assays.
Although significant biological evaluation has been carried out on Hp-MAP3, I have the following concerns on the manuscript:
1. In the introduction section, the authors mentioned that it is important to develop new drug therapies against cancer cells. However, there is only minimal background information regarding cancer in the introduction section of the manuscript.
2. In section 2.1, the authors provided the hydrophobicity, portion of the hydrophobic residues and hydrophobic moment of Hp-MAP3. The authors should mention how these data were calculated.
3. Please define what “Nd” means in Table 1. Is there any reason why the MBCs of ciprofloxacin against the two strains were not determined?
4. The authors should consider testing Hp-MAP3 against antibiotic-resistant strains.
5. In line 123 of section 2.3, “MIC” should be “MBC”
6. Figure 3,
a. y-axis label should be “Biofilm growth” or “Biofilm mass”.
b. I recommend the authors to change the x-axis labels to “0.5×MIC”, “1×MIC” etc. to allow readers easily correlating the percentage biofilm mass with the MIC of Hp-MAPs. This should also apply to Figure 5.
c. It is not surprised that nearly 100% inhibition of biofilm growth was observed at or above 1×MIC of Hp-MAP3 due to the bactericidal effect of Hp-MAP3 at such concentrations. The authors should focus on testing the ability of Hp-MAP3 to inhibit biofilm formation at sub-MIC concentrations. Hp-MAP3 was only tested at 0.5×MIC (and above). The authors should test it at lower concentrations (such as 0.25×MIC and 0.125×MIC)
7. A significant portion of chronic bacterial infections involve established biofilms. Was Hp-MAP3 tested for its ability to eradicate pre-established biofilms?
8. Hp-MAP3 was only tested against one cancer cell line (HeLa). More biological evaluation on the anticancer properties of Hp-MAP3 (such as cytotoxicity against different cell lines and mechanistic studies) should be carried out.
9. The IC50 of Hp-MAP3 against MRC-5 is 19.8 uM. The selective indices of Hp-MAP3 for all but one of the microbial strains are below 4, suggesting a low selectivity and narrow therapeutic window for the compound. This is further confirmed by the hemolytic assay. The clinical application of Hp-MAP3 will be significantly impeded by its relatively high cytotoxicity. The authors should attempt modifying the structure of Hp-MAP3 to reduce the cytotoxicity of the compound.
10. The authors should provide the chromatogram of Hp-MAP3 as evidence of the purity of the compound.
11. There are some spelling, grammatical, punctuational and linguistic mistakes throughout the manuscript. The authors should read the manuscript carefully and fix all these mistakes.
Author Response
Review 3:
1) In the introduction section, the authors mentioned that it is important to develop new drug therapies against cancer cells. However, there is only minimal background information regarding cancer in the introduction section of the manuscript.
Response: The authors agree and the topic of tumor cells were added.
Now reads:
(Introduction: Page 2, line 53):
“According to a report by the World Health Organization, cancer is responsible for about 10 million deaths worldwide by 2020 [11]. Conventional treatments involving surgery, chemotherapy and radiotherapy, along with newly developed immunotherapy, have been applied to eliminate cancer cells or inhibit their proliferation [12]. Lung cancer is the leading cause of cancer death worldwide, with 2.21 million cases [13]. Persistent high-risk HPV infections is the main etiological factor of cervical cancer with 1.4 million cases worldwide [14–16]. This type of cancer can lead to changes in immunity and in the vaginal microbiota, leaving the patient vulnerable to resistant bacterial and fungal infections [17]. Rhabdomyosarcoma (RD) soft tissue sarcoma is responsible for ~7% of cancers in children. They spread mainly sporadically (90%) and are therefore rarely associated with an underlying constitutional genetic disease (10%). It represents a high-grade neoplasm of skeletal cells similar to myoblasts. Treatment is invasive by surgical resection and/or ionizing radiation and eradication of systemic metastatic disease with intensive chemotherapy [18–20].
- In section 2.1, the authors provided the hydrophobicity, portion of the hydrophobic residues and hydrophobic moment of Hp-MAP3. The authors should mention how these data were calculated.
Response: The authors agree and the results were better described.
Now reads:
(Results: Page 3, line 102):
“Features calculated on the HeliQuest website according to the Eisenberg scale”.
- Please define what “Nd” means in Table 1. Is there any reason why the MBCs of ciprofloxacin against the two strains were not determined?
Response: The authors agree with the reviewer and the term Nd was removed. Now, the Table 1 updated is represented by maximum concentration assayed for each bacterium (e.g. >100.8 µM). Ciprofloxacin was the unique antibiotic active against all clinical isolated bacteria tested including the β-lactams. In addition, the mechanism of action for ciprofloxacin is on DNA gyrase inactivation, blocking the bacterial metabolism.
- The authors should consider testing Hp-MAP3 against antibiotic-resistant strains.
Response: The authors agree and the assays realized with the clinical isolate bacteria were initially evaluated as ATCC however the antibiogram demonstrated that the clinical isolates were resistant strains for all antibiotic tested with exception of ciprofloxacin due to mechanism of action. The antibiogram is presented in Table S1.
- In line 123 of section 2.3, “MIC” should be “MBC”
Response: The authors agree and the modification was realized.
- Figure 3,
- y-axis label should be “Biofilm growth” or “Biofilm mass”.
Response: After observation realized by referee the authors modified the Figure 3 y-axis to “Biofilm mass”.
- I recommend the authors to change the x-axis labels to “0.5×MIC”, “1×MIC” etc. to allow readers easily correlating the percentage biofilm mass with the MIC of Hp-MAPs. This should also apply to Figure 5.
Response: The authors agree and the idea was added in both Figure 3 and Figure 5.
- It is not surprised that nearly 100% inhibition of biofilm growth was observed at or above 1×MIC of Hp-MAP3 due to the bactericidal effect of Hp-MAP3 at such concentrations. The authors should focus on testing the ability of Hp-MAP3 to inhibit biofilm formation at sub-MIC concentrations. Hp-MAP3 was only tested at 0.5×MIC (and above). The authors should test it at lower concentrations (such as 0.25×MIC and 0.125×MIC).
Response: The authors agree. Information about MIC values such as 0.5×; 1×; 2×; 4×; 8× and 16×MIC were added in Figures 4 and 5. Results for 0.5×MIC activity were not observed and therefore the concentration below the sub-MICs were not tested.
- A significant portion of chronic bacterial infections involve established biofilms. Was Hp-MAP3 tested for its ability to eradicate pre-established biofilms?
Response: The authors agree, however, that the ability to eradicate pre-established biofilms has not yet been observed due to standardization tests in our group. The authors can carry out the experiment if necessary within three weeks.
- Hp-MAP3 was only tested against one cancer cell line (HeLa). More biological evaluation on the anticancer properties of Hp-MAP3 (such as cytotoxicity against different cell lines and mechanistic studies) should be carried out.
Response: The authors agree and results with other tumor cell lines were also added. Regarding mechanistic studies, the idea is excellent, but we decided to leave it to write an in-depth article only on this subject.
Now reads:
(Results: Page 6, line 167):
“The cytotoxic activity of the new Hp-MAP3 analog peptide against healthy MRC-5 cell lines and NCI-H292, RD, and HeLa tumor cells. The results showed that Hp-MAP3 inhibited 51% of the cellular viability of RD, 65% of the NCI-H292 line, and 85% of the HeLa line, all at the concentration of 21.6 μM (Figure 4). The value of IC50 was determined using the dose-response curve. Hp-MAP3 presented an IC50 for the tested lines MRC-5, RD, NCI-H292, and HeLa of 19.8, 24.0, 15.4, and 10.4 μM, respectively. The calculation of the selectivity index (IS) of the peptide for the tumor cell lines, RD, NCI-H292, and HeLa was 0.8, 1.2, and 1.8, respectively (Table 3)”.
(Discussion: Page 9, line 284):
“AMPs have shown promising results as new therapeutic agents in cancer, as they are membrane selective [66]. Brevinin-2R is an antimicrobial peptide from the cutaneous secretion of Rana ridibunda that has a membrane-disruptive mode of action. The results demonstrated that brevinine inhibited the proliferation of 50% of HeLa cells at ~80 μM [67]. Another example is aurein peptides 1.2 and 3.1 from Litoria sp. that showed activity against the NCI cancer cell line, with LC50 values of 4 and 10 μM, respectively [68]. Furthermore, data on RD cells and peptides are scarce in the literature, but recent studies have shown that phospholipase (BmPLA2) from Bothrops moojeni demonstrated an IC50 of ~0.14 μM [69].”
- The IC50 of Hp-MAP3 against MRC-5 is 19.8 uM. The selective indices of Hp-MAP3 for all but one of the microbial strains are below 4, suggesting a low selectivity and narrow therapeutic window for the compound. This is further confirmed by the hemolytic assay. The clinical application of Hp-MAP3 will be significantly impeded by its relatively high cytotoxicity. The authors should attempt modifying the structure of Hp-MAP3 to reduce the cytotoxicity of the compound.
Response: The observation made by the reviewer is relevant and the authors agree. Our research group has been working overtime on the construction of new toxin-based analogues. Therefore, we sometimes arrive at peptides with promising activities, but also very cytotoxic. These obstacles in construction offer us more and more subsidies to understand the structure-function relationship and create new rationales for the construction of the next generations. Furthermore, the use of these candidates combined with nanotechnology and the utopian use for example, it might be an outlet to use these candidates.
- The authors should provide the chromatogram of Hp-MAP3 as evidence of the purity of the compound.
Response: The authors agree and the supplemental material was added in the result section.
- There are some spelling, grammatical, punctuational and linguistic mistakes throughout the manuscript. The authors should read the manuscript carefully and fix all these mistakes.
Response: The authors agree and all manuscript was revised.

Round 2
Reviewer 3 Report
The authors have addressed most of the issues raised by the reviewers and improved the manuscript. However, the following aspects should be addressed before it can be considered for publication.
1. In Figure 3,
a. The biofilm mass of the negative control (A. baumannii) should be 100%.
b. In line 155, the authors mentioned that “at 0.5× MIC it inhibited ~42% of the biofilm mass”, however, figure 3 indicates ~60% inhibition at that concentration.
c. In response to reviewer 3’s comment, the authors mentioned that “Results for 0.5×MIC activity were not observed and therefore the concentration below the sub-MICs were not tested.” However, figure 3 indicates ~60% biofilm inhibition at 0.5×MIC of Hp-MAP3, suggesting that the compound had good biofilm inhibition at such concentration. The authors should consider testing the compound at lower concentration (such as 0.25×MIC and 0.125×MIC).
2. As pointed out by reviewer 3 that a significant portion of chronic bacterial infections involve established biofilms. The authors should test Hp-MAP3 for its ability to eradicate pre-established biofilms in biofilm disruption assay.
3. In figure 4, the error bar for Triton (1%) is missing.
4. In lines 233-234, the authors claimed that “Hp-MAP3, showed MIC ranging from 2.7 to 43.3 μM, demonstrating a higher antimicrobial activity”. However, in lines 232-233, the author mentioned that “the two 231 analogs Hp-MAP1 and Hp-MAP2 were actives also against bacteria Gram-negative and 232 -positive with MIC values of 2 to 45 μM”, suggesting that Hp-MAP3 only had comparable antimicrobial activity compared to Hp-MAP1 and Hp-MAP2.
5. There are still some grammatical mistakes in the manuscript (especially in the revised parts). The authors should read the manuscript carefully again and fix all these mistakes.
Author Response
Editor Office
Toxins
Manuscript reference number: 1884032
Please find attached the revised version of our manuscript entitled “Evaluation of the synthetic multifunctional peptide Hp-MAP3 inspired by temporin-PTa". The comments of the reviewers were enlightening, improving the quality of our paper. In the next pages, are our present the changes suggested by the reviewers.
Comments to the Author
Review 3:
- In Figure 3,
- The biofilm mass of the negative control (A. baumannii) should be 100%.
Response: The authors agree and Figure 3 was changed.
- In line 155, the authors mentioned that “at 0.5× MIC it inhibited ~42% of the biofilm mass”, however, figure 3 indicates ~60% inhibition at that concentration.
Response: The authors agree and the change was made to the text.
- In response to reviewer 3’s comment, the authors mentioned that “Results for 0.5×MIC activity were not observed and therefore the concentration below the sub-MICs were not tested.” However, figure 3 indicates ~60% biofilm inhibition at 0.5×MIC of Hp-MAP3, suggesting that the compound had good biofilm inhibition at such concentration. The authors should consider testing the compound at lower concentration (such as 0.25×MIC and 0.125×MIC).
Response: The authors agree and if it is really necessary, the correction period needs to be longer, as the test takes up to two weeks to be performed.
- As pointed out by reviewer 3 that a significant portion of chronic bacterial infections involve established biofilms. The authors should test Hp-MAP3 for its ability to eradicate pre-established biofilms in biofilm disruption assay.
Response: The authors agree and if it is really necessary, the correction period needs to be longer, as the test takes up to two weeks to be performed.
- In figure 4, the error bar for Triton (1%) is missing.
Response: The authors agree and standard deviation was added in the Figure 4.
- In lines 233-234, the authors claimed that “Hp-MAP3, showed MIC ranging from 2.7 to 43.3 μM, demonstrating a higher antimicrobial activity”. However, in lines 232-233, the author mentioned that “the two 231 analogs Hp-MAP1 and Hp-MAP2 were actives also against bacteria Gram-negative and 232 -positive with MIC values of 2 to 45 μM”, suggesting that Hp-MAP3 only had comparable antimicrobial activity compared to Hp-MAP1 and Hp-MAP2.
Response: The authors agree and the sentence was modified.
- There are still some grammatical mistakes in the manuscript (especially in the revised parts). The authors should read the manuscript carefully again and fix all these mistakes.
Response: The authors agree and the entire text was carefully revised.
